# Yeast Culture Supplementation Improves Meat Quality by Enhancing Immune Response and Purine Metabolism of Small-Tail Han Sheep (*Ovis aries*)

**DOI:** 10.3390/ijms26104512

**Published:** 2025-05-09

**Authors:** Xiaobo Bai, Liwei Wang, Hua Sun, Lvhui Sun, Jianghong An, Shaoyin Fu, Mengran Zhao, Fang Liu, Xiaoqi Ren, Zheng Liu, Jiangfeng He, Yongbin Liu

**Affiliations:** 1College of Life Sciences, Inner Mongolia University, Hohhot 010030, China; 32208159@mail.imu.edu.cn (X.B.); 18747977760@163.com (L.W.); 2Inner Mongolia Academy of Agricultural & Animal Husbandry Sciences, Hohhot 010031, China; sunhua564116908@163.com (H.S.); ajh_gz@163.com (J.A.); fushao1234@126.com (S.F.); burutiaowuyjd@163.com (M.Z.); liuf185@163.com (F.L.); 3State Key Laboratory of Agricultural Microbiology, Hubei Hongshan Laboratory, Frontiers Science Center for Animal Breeding and Sustainable Production, College of Animal Science and Technology, Huazhong Agricultural University, Wuhan 430070, China; lvhuisun@mail.hzau.edu.cn; 4College of Animal Science and Technology, Inner Mongolia Minzu University, Tongliao 028000, China; 18747599252@139.com (X.R.); lz010625@163.com (Z.L.); 5College of Animal Science, Inner Mongolia Agriculture University, Hohhot 010018, China

**Keywords:** yeast culture, immune response, purine metabolism, meat quality, small-tail Han sheep

## Abstract

Yeast culture is widely used in ruminants to improve gut health, immunity, and productivity; however, its impact on meat quality remains unclear. This study aimed to investigate the effects of yeast culture supplementation in the basic diet on meat quality of Small-tail Han sheep. A total of 40 Small-tail Han sheep (17.5 ± 1.2 kg) were randomly assigned to two treatment groups, with 20 sheep in each group. The sheep were fed either a basic diet (CON) or the basic diet supplemented with 1% yeast culture (YSD) for 90 days. At the end of the trial, the Longissimus dorsi muscle (LOD) of the sheep was collected for meat quality evaluation, as well as transcriptome and metabolome analyses. Meat quality data were analyzed using *t*-tests, while transcriptome and metabolome data were analyzed using bioinformatics tools. The results showed that YSD supplementation significantly reduced carcass fat content (*p* < 0.05) and increased the pH values (*p* < 0.05) of LOD compared to the CON group. Multi-omics analysis revealed significant changes in the levels of 349 transcripts and 149 metabolites (*p* < 0.05) in the YSD group relative to the CON group. These changes were primarily associated with immune response pathways and purine metabolism. Further integrated transcriptomics and metabolomics analysis identified significant alterations in the expression of adenylate kinase 4 (*AK4*) and ribonucleotide reductase M2 (*RRM2*), which influenced purine metabolites, such as ADP, GMP, 3′-AMP, 3′-GMP, dGDP, adenine, guanosine, and guanine. These metabolites were markedly upregulated in the LOD of the sheep supplemented with yeast culture. In conclusion, yeast culture supplementation improved the meat quality of Small-tail Han sheep, potentially through the enhancement of immune response and purine metabolism. These findings offer valuable insights into the molecular mechanisms underlying the effects of yeast culture on animal health and meat quality.

## 1. Introduction

In recent years, the global livestock industry has experienced significant growth. Statistics indicate that global meat consumption has steadily increased over the past few decades, with total meat consumption projected to double by 2050 compared to 2010 [1]. Among various meat products, mutton stands out as a high-quality source of animal protein, providing essential amino acids, vitamins (such as vitamin B), and minerals (including iron, zinc, and phosphorus) [2]. Furthermore, the rising global population and growing consumer awareness of healthy eating habits have contributed to the increasing demand for mutton in recent years.

The small-tailed Han sheep (*Ovis aries*) is a unique mutton sheep breed native to China, primarily distributed in Inner Mongolia, Gansu, and Ningxia. Renowned for its rapid growth, high-quality meat, and strong adaptability, this breed plays a vital role in meat production in northern China [3]. Its meat is highly favored by consumers for its delicious taste, rich nutritional value, and low-fat content, making it a competitive product in regional meat markets. With the continuous expansion of the mutton market, the breeding scale of Small-tailed Han sheep has been steadily increasing, particularly through intensive farming practices.

Under intensive production systems, high-grain feeds are widely used to enhance the growth performance and feed efficiency of sheep, thereby achieving greater economic returns [4,5]. While this feeding strategy has significantly improved production efficiency, it has also raised concerns about increased risk of metabolic disorders [6]. One major issue associated with high-grain feeding is the abnormal accumulation of organic acids in the rumen, which disrupts the microbial balance [7]. Specifically, this imbalance leads to the massive death of gram-negative bacteria, resulting in the release of endotoxins. These endotoxins can trigger a series of inflammatory responses, ultimately compromising the health and well-being of ruminants [8].

Yeast culture is a microbial product created through yeast fermentation under specific conditions on a particular medium [9]. It contains over 100 types of metabolic products, including vitamins, minerals, digestive enzymes, organic acids, oligosaccharides, peptides, and amino acids [10]. As a green additive, yeast culture is widely used in ruminant production for its beneficial effects on improving production performance and animal health [11]. For instance, the addition of yeast culture to the diet of Egyptian buffaloes improved feed conversion rate, energy utilization, and nitrogen efficiency [12]. Moreover, supplementing sheep diets with yeast culture has been shown to alleviate acidosis caused by high-grain feed in sheep [13]. Therefore, yeast culture supplementation can significantly enhance animal production performance and health.

With the advancement of omics technologies, researchers have increasingly focused on identifying compounds that influence meat quality through metabolomics, providing a scientific foundation for its improvement [14]. Studies have shown that various compounds in meat can directly or indirectly affect its quality and flavor [15,16]. In lamb, specific free amino acids (e.g., glutamate, aspartate, alanine) and nucleotides (e.g., 5′-inosine monophosphate [IMP], adenosine 5′-monophosphate [AMP], and 5′-guanosine monophosphate [GMP]) are recognized as key contributors to the umami taste [17,18]. Moreover, integrated transcriptomics and metabolomics approaches have proven valuable in elucidating the molecular mechanisms underlying the effects of yeast culture supplementation on meat quality. While previous studies have examined the impact of yeast culture on the growth performance and rumen microbiota of sheep [19,20], limited research has investigated its influence on muscle metabolites and meat flavor. To address this gap, this study explores the effects of yeast culture supplementation in the diet on slaughter performance, muscle metabolites, and meat quality of Small-tail Han sheep.

## 2. Results

### 2.1. Slaughter Performance

The effects of yeast culture supplementation on the slaughter performance of Small-tail Han sheep are illustrated in Figure 1. There was no significant difference in final body weight (*p* > 0.05, Figure 1A) or carcass weight (*p* > 0.05, Figure 1B) between the CON group and YSD groups at the end of the feeding period. However, the Girth rib (GR) value in the YSD group was significantly reduced by 22.8% compared to the CON group (*p* < 0.05, Figure 1C). Additionally, the pH value of the LOD in the YSD group increased significantly by 2.1% (*p* < 0.01, Figure 1D).

### 2.2. Transcriptomic Analysis in the LOD Muscle

A total of 17,082 transcripts were identified in the LOD through mRNA sequence analysis. Among these, 286 genes were upregulated and 63 were downregulated in the YSD group relative to the CON group (Figure 2A). Gene Ontology (GO) enrichment analysis revealed 106 significantly enriched clusters (q < 0.001) in the YSD group compared to the CON group. The top 10 terms were mainly involved in biological processes (BP), cellular components (CC), and molecular functions (MF), such as positive regulation of interferon-gamma secretion, immune response, regulation of adaptive immune response, and external side of plasma membrane, antigen binding, and signaling receptor activity (Figure 2B).

Additionally, KEGG pathway analysis identified significant enrichment in 60 pathways among the differentially expressed genes (DEGs) in the YSD group compared to the CON group. The top 10 pathways were associated with natural killer cell-mediated cytotoxicity, complement and coagulation cascades, staphylococcus aureus infection, and the chemokine signaling pathway (Figure 2C). These results align with the GO enrichment analysis, highlighting the intricate interplay between immune response and pathogen–host interactions.

Focusing on the natural killer cell-mediated cytotoxicity pathway (oas04650), genes were selected for further investigation based on enrichment degree and the number of significantly enriched genes. Within this pathway, 12 DEGs were identified. Notably, upregulated genes included *CD48*, *CD94*, *NKG2D*, *2B4*, *LCK*, *LCP2*, *ZAP70*, *VAV*, *SLP-76*, *DAP-12,* and *Granzyme*, while the downregulated gene was *JCAM1/2* in the YSD group compared to the CON group. The expression of these genes in both groups is shown in Figure 2D. Furthermore, the molecular interactions and signaling cascades related to the cytotoxic activity of natural killer cells are shown in Figure 2E, illustrating the complex interactions within this pathway.

### 2.3. Metabolomic Analysis in the LOD Muscle

Metabolomic analysis was conducted to investigate the effects of yeast culture supplementation on meat quality. Orthogonal Partial Least Squares Discriminant Analysis (OPLS-DA) revealed clear segregation and significant differences between the CON and YSD groups (Figure 3A). Using the variable importance in projection (VIP > 1) from the first principal component of the OPLS-DA and the *p*-value from the *t*-test (*p* < 0.05), a total of 6414 metabolites were identified in the LOD. Among these, 103 differential metabolites (DMs) were significantly upregulated, while 46 DMs were significantly downregulated in the YSD group compared to the CON group (Figure 3B).

The identified metabolites included 30 organic acids and their derivatives, 25 lipids and lipid-like molecules, 16 organic heterocyclic compounds, 11 organic oxygen-containing compounds, 9 nucleotides and their analogs, 8 phenylpropanoids and ketones, 3 phenyl ring-containing compounds, 1 organic nitrogen compound, 1 alkaloid and its derivatives, and 45 unclassified compounds (Figure 3C). The compounds highlighted the metabolic differences between the YSD and CON groups. KEGG pathway analysis indicated that the DMs were primarily involved in purine metabolism, pantothenate and CoA biosynthesis, vitamin digestion and absorption, riboflavin metabolism, thermogenesis, taste transduction, FoxO signaling pathway, and lysosomal pathway (Figure 3D).

Notably, purine metabolites such as ADP, GMP, 3′-AMP, 3′-GMP, dGDP, adenine, guanosine, and guanine were significantly elevated in the YSD group, with levels ranging from 1.27- to 2.03-fold higher compared to the CON group (Figure 3E). Among other DMs, the content of inositol 1,3,4,5,6-pentakisphosphate (IP5) in the phosphatidylinositol signaling system was 4.56 times higher in the YSD group than in the CON group. Similarly, pantetheine 4′-phosphate, a key metabolite in both the pantothenate biosynthesis pathway and the CoA biosynthesis pathway, showed a 1.75-fold increase in the YSD group. In contrast, the level of panthenol was 0.79 times lower in the YSD group compared to the CON group (Figure 3F).

### 2.4. Combined Transcriptomic and Metabolomic Analysis

Correlation analysis of the top 100 DEGs and top 100 DMs is presented in Figure 4A. Among these, *AK4* and *RRM2* were identified as key genes primarily involved in purine metabolism. Notably, *AK4* was significantly downregulated, while *RRM2* was significantly upregulated in the YSD group compared to the CON group. Additionally, *AK4* also plays a role in tryptophan metabolism, whereas *RRM2* is involved not only in purine metabolism but also in other metabolic pathways, including pyrimidine metabolism, glutathione metabolism, and drug metabolism involving other enzymes. Based on these findings, *AK4* and *RRM2* were selected for further analysis in relation to the DMs that were significantly upregulated in purine metabolism (Figure 4B). Specifically, dGDP was identified as being primarily by *RRM2*. Furthermore, *AK4* and *RRM2* may influence ADP production and conversion through an antagonistic mechanism, suggesting a complex regulatory interplay between these two genes.

## 3. Discussion

Yeast culture is widely believed to enhance ruminant performance by stimulating the growth of fiber-digesting microorganisms [21,22,23]. However, studies on its effects have yielded inconsistent results. While some research indicates that yeast culture supplementation improves feed intake and body weight in ruminants [24,25], other studies report no significant effects on production performance when yeast or its culture is added to feed [26,27]. In this study, no significant differences were observed in the final body weight or carcass weight of Small-tail Han sheep between the CON and YSD groups. However, the YSD group exhibited a significantly lower GR value compared to the CON group, without any weight loss after 90 days of feeding. This suggests that yeast culture may effectively reduce fat deposition in Small-tail Han sheep without compromising their weight, which is valuable for production purposes.

The fat content in meat also impacts the fatty acid composition. An increase in fat content tends to elevate the levels of monounsaturated fatty acid (MUFA) and saturated fatty acid (SFA) more rapidly than polyunsaturated fatty acid (PUFA), thereby reducing the ratio of unsaturated fatty acid (USFA) to SFA (U/S). This imbalance poses potential risks to human health [28]. Our findings suggest that supplementing the diet of Small-tail Han sheep with 1% yeast culture can effectively reduce fat deposition while maintaining body weight. This provides new insights into the role of yeast culture in improving animal health and meat quality.

Excessive fat deposition, often caused by intensive feeding and high concentrate input, negatively impacts feed utilization and meat quality, affecting attributes such as color, pH, tenderness, and flavor [29]. Additionally, the accumulation of lactic acid can lower pH of meat, increasing the risk of pale, soft, exudative (PSE) meat. In this study, the pH value of the LOD was significantly higher in the YSD group after supplementation with 1% yeast culture [30]. This indicates that yeast culture supplementation may reduce lactic acid accumulation in the LOD and lower the incidence of PSE meat, contributing to improved meat quality.

To better understand how yeast culture enhances productivity and meat quality in sheep, we employed an innovative approach combining comprehensive transcriptomics and metabolomics analyses to investigate the effects of dietary yeast culture on Small-tail Han sheep. Transcriptome analysis revealed that 349 DEGs affected by yeast culture supplementation were mainly enriched in biological process, including immune response, regulation of adaptive immune response, antigen binding, signaling receptor activity, natural killer cell mediated cytotoxicity, and chemokine signaling pathway. A balanced immune system is critical for maintaining animal health [31]. In this study, KEGG enrichment analysis showed that the natural killer (NK) cell-mediated cytotoxicity pathway was significantly altered in the YSD group compared to the CON group, with key immune-related genes such as *CD48*, *CD94*, *NKG2D*, *2B4*, *LCK*, *LCP2*, *ZAP70*, *VAV*, *SLP-76*, *DAP-12*, *Granzyme* being upregulated. Among these genes, 2B4 plays a crucial role in regulating the activity of NK cells, T cells and other immune cells. It modulates immune responses by binding to *CD48* and interacts with SAP proteins via immunoreceptor tyrosine-based switch motifs (ITSMs) in its intracellular structural domains, transmitting activating or inhibitory signals [32]. *NKG2DL* serves as a ligand for the *NKG2D* receptor, an activating receptor expressed on T cells. This receptor recognizes atypical MHC class I molecules expressed during viral infections or certain tumor cells, thereby activating the cytotoxic functions of NK cells and T cells [33]. In contrast, *CD94* forms heterodimers with members of the NKG2 family, such as *NKG2A* and *NKG2C*, to create receptors that bind to MHC class I molecules and regulate the activation or inhibition of NK cells [34]. Granzyme, a serine protease released by NK cells and cytotoxic T cells, induces apoptosis in target cells and plays a critical role in immune cell-mediated cytotoxicity [35]. Therefore, dietary yeast culture supplementation upregulated the NK cell-mediated cytotoxicity pathway by modulating the expression of immune-related genes, thereby enhancing the immune function of Small-tail Han sheep. This finding suggests that yeast culture may play a significant role in improving the health and productivity of sheep through immune system regulation.

Metabolome analysis revealed that 149 DMs were affected by yeast culture supplementation. KEGG pathway analysis showed that these DMs were mainly involved in purine metabolism, FoxO signaling pathway, vitamin digestion and absorption, and taste transduction. Among these, dietary supplementation with 1% yeast culture significantly increased the levels of purine metabolites, including ADP, GMP, 3′-AMP, 3′-GMP, dGDP, adenine, guanosine, and guanine. Purine compounds are widely recognized for their association with umami taste, which is often stronger in meat with higher purine content [36]. The role of purine metabolism in improving umami taste has been extensively studied. For instance, research on pork has shown that guanine content is positively correlated with umami flavor [37]. Similarly, studies on chicken have highlighted that purine compounds, particularly inosine monophosphate (IMP), can significantly improve umami taste [38]. Among the purine metabolites, GMP is a well-known umami enhancer. When combined with sodium glutamate, GMP exhibits a remarkable synergistic effect, amplifying the umami taste by several to tens of times. Furthermore, guanine also shows a significant positive correlation with umami taste, and GMP and guanine can interconvert within the purine metabolism pathway [37]. In this study, the addition of 1% yeast culture in the diet likely enhanced the umami flavor through the synergistic interaction between GMP and guanine. Additionally, GMP-related metabolites such as dGDP, 3′-GMP, and guanosine were notably elevated in the YSD group compared to the CON group, collectively contributing to improved muscle umami. B vitamins play a critical role in improving the productivity, reproductive performance, and immune function of ruminants [39]. This study found that yeast culture supplementation significantly enhanced the absorption and metabolism of vitamins, particularly vitamin B2 and B5. It also promoted the biosynthesis of pantothenic acid and CoA, and improved metabolic pathways in the LOD. These findings suggest that yeast culture supplementation not only enhances immunity, as supported by transcriptomic results, but also contributes to overall animal health and meat quality. Therefore, dietary yeast culture is essential for improving the health and meat quality of sheep.

The gene–metabolite interaction network revealed that *RRM2*, a gene associated with purine metabolism, was significantly upregulated, while adenylate kinase *AK4* was notably downregulated in the YSD group compared to the CON group. *RRM2*, a minor subunit of ribonucleotide reductase, plays a critical role in nucleotide metabolism by facilitating the conversion of ribonucleotides to deoxyribonucleotides. This process sustains the dNTP pool essential for DNA synthesis, repair, and replication [40]. In contrast, *AK4,* which is predominantly localized in the mitochondrial matrix, is involved in cellular energy metabolism and the regulation of intracellular ATP levels. Additionally, *AK4* contributes to nucleotide balance and energy homeostasis by catalyzing phosphorylation group transfer reactions among nucleotides [41,42]. Interestingly, *RRM2* and *AK4* exhibit opposing roles in purine metabolism. *RRM2* primarily supports nucleotide biosynthesis, while *AK4* regulates energy metabolism and intracellular ATP levels. In this study, we found that dietary supplementation with 1% yeast culture may influence purine metabolism by modulating ADP production and conversion through the antagonistic interaction between *RRM2* and *AK4.* This regulatory mechanism may contribute to improved lamb flavor by enhancing purine-related metabolites.

## 4. Materials and Methods

### 4.1. Experimental Design and Sample Collection

The animal experimental protocol was approved by the Institutional Animal Care and Use Committee of the Inner Mongolia Academy of Agriculture and Animal Husbandry Science (Hohhot, China; approval no. 2022003). A total of 40 Small-tail Han sheep (*Ovis aries*) weighing 17.5 ± 1.2 kg were randomly assigned to two treatment groups, with each group comprising 20 sheep. The sheep were fed either a basic diet (CON) or the basic diet supplemented with 1% yeast culture (YSD). Based on relevant literature, supplementation with 1% yeast culture has been shown to improve corn stover dry matter disappearance (DM), neutral detergent fiber digestibility (NDF), and forage fiber degradation. Therefore, we selected 1% yeast culture for supplementation in accordance with the product guidelines [43]. The ingredients and nutritional composition of the experimental diets are detailed in Table 1.

During the experiment, all sheep were housed in a standardized sheep facility. Each group was kept in separate pens to ensure adequate space for movement and activity. The sheep house was well-ventilated, with temperature and humidity maintained within suitable ranges to ensure the comfort of the animals. The facility was cleaned daily to maintain hygiene and prevent the occurrence and spread of diseases. Throughout the trial, the lambs had unrestricted access to clean drinking water and salt blocks, ensuring their nutritional and hydration needs were met.

At the end of the 90-day feeding period, 10 lambs from each group were deprived of food and water for 24 h prior to slaughter. The slaughter process adhered strictly to ethical regulations, and professionally trained personnel performed the procedure to minimize pain and stress. Following slaughter, samples of the longissimus dorsi (LOD) muscle were collected immediately from the left side of the 12th and 13th ribs. These samples were labeled, snap-frozen in liquid nitrogen, and stored at −80 °C for subsequent analysis.

### 4.2. Carcass Characteristics and Meat Quality Analysis

After the feeding experiment, 10 lambs from each treatment group were randomly selected and individually weighed after a 24 h fasting period. The carcass weight was measured using an electronic scale with an accuracy of 0.1 kg immediately after slaughter. During the skinning process, careful operation was performed to prevent damage to the muscle tissue. When removing internal organs, including the stomach, intestine, liver, spleen, and other organs, the kidney and perirenal fat were retained. The girth rib (GR) value, measured 110 mm away from the center of the 12th and 13th ribs, was also measured; it was recorded to represent the carcass fat content. The pH value was measured within 45 min after slaughter using a portable pH meter. Before measurement, the electrode of the pH meter was cleaned and calibrated using standard buffers (pH 4.0 and 7.0) to ensure the accuracy of the results. The electrodes were inserted 2–3 cm into the sheep LOD and the measurements were repeated three times and averaged.

### 4.3. Total RNA Extraction and Transcriptomic Analysis

In this study, transcriptome sequencing was performed on LOD samples from Small-Tail Han sheep. To ensure that each sample accurately represented the gene expression of an individual sheep, independent samples were collected from the LOD of each animal. Immediately after collection, the samples were labeled, snap-frozen in liquid nitrogen, and stored at −80 °C to preserve RNA integrity for subsequent transcriptome analysis.

#### 4.3.1. mRNA Enrichment and Sequencing

Total RNA was extracted from LOD tissue using the TRIzol reagent kit (Invitrogen, Carlsbad, CA, USA) according to the manufacturer’s protocol. The purity, quantification, and integrity of the RNA were evaluated using a NanoDrop 2000 Spectrophotometer (Thermo Scientific, Waltham, MA, USA) and an Agilent 2100 Bioanalyzer (Agilent Technologies, Santa Clara, CA, USA). The samples with RNA Integrity Number (RIN) ≥ 7 were selected for further analysis to ensure high-quality RNA for downstream applications. Library preparation was performed using the TruSeq Stranded mRNA LT Sample Prep Kit (Illumina, San Diego, CA, USA) according to the manufacturer’s instructions. The prepared libraries were sequenced on the Illumina platform, generating 125 bp/150 bp paired-end reads for transcriptome analysis.

#### 4.3.2. Quality Control and Sequence Mapping

The raw sequencing reads were processed using Trimmomatic software (version 0.36, Bjorn Usadel’s group at RWTH Aachen university, Aachen, Germany) to ensure the data quality. During this step, reads containing poly-N sequences and those with low-quality scores were removed, producing a clean set of high-quality reads. Following quality control, the refined reads were aligned to the reference genome using the HISAT2 alignment tool, which ensures accurate and efficient mapping of sequencing data to the genome.

#### 4.3.3. Gene Expression Quantification and Differential Expression Analysis

Differentially expressed genes (DEGs) were identified using the R (version 4.1.2)’s DESeq R package, which provides tools for size factor estimation and statistical testing. To ensure robust results, a stringent threshold was applied; genes with a *p*-value < 0.05 and a fold change greater >2 or <0.5 were considered significantly differentially expressed. The expression patterns of these DEGs were further explored using hierarchical clustering to visualize similarities and differences among samples. To gain biological insights, functional enrichment analyses for Gene Ontology (GO) and Kyoto Encyclopedia of Genes and Genomes (KEGG) pathways were performed in R, utilizing the hypergeometric distribution model to identify significantly enriched categories.

### 4.4. Metabolite Extraction and Metabolomic Analysis

#### 4.4.1. Metabolite Extraction

The LOD tissue (30 mg) was accurately weighed into a 1.5 mL EP tube. To each sample, 20 μL of internal standard solution (L-2-chlorophenyl alanine, 0.3 mg/mL; Lyso PC17:0, 0.01 mg/mL, both prepared in methanol) and 400 μL of methanol–water (*v*:*v* = 4:1) were added to 30 mg samples. The mixture was vigorously agitated for 60 s, followed by ultrasonic treatment for 10 min. Afterward, the samples were stored at −20 °C for 30 min. The homogenate extract was then centrifuged at 16,173× *g* at 4 °C for 15 min. The resulting supernatants (300 μL) were dried using a freeze-concentration centrifugal dryer and stored in a brown glass vial to prevent light-induced degradation. Next, 200 μL of solvent mixture (methanol–water, *v*:*v* = 1:4) was added to the dried extract, and the solution was passed through a 0.22 μm filter before being transferred into liquid chromatography vials. The filtrate was stored at −80 °C for subsequent LC-MS/MS analysis. To ensure the stability of the LC-MS/MS analysis process, quality control (QC) samples were prepared by pooling equal volumes from all individual samples.

#### 4.4.2. UHPLC-QE-MS Analysis and Data Preprocessing

For metabolomics profiling, separations were performed using an ACQUITY UPLC HSS T3 column (100 mm × 2.1 mm, 1.8 μm particles, Waters (Waters Corporation, Milford, Milford, CT, USA) maintained at 45 °C. The flow rate was set at 0.35 mL/min, and the injection volume was 2 μL. A binary gradient elution system was employed, consisting of 0.1% formic acid in water (solvent A) and 0.1% formic acid in acetonitrile (solvent B). The gradient elution program was as follows: 0–2 min, 5% B; 2–4 min, 5–25% B; 4–8 min, 25–50% B; 8–10 min 50–80% B; 10–14 min, 80–100% B; 14–15 min, 100% B; 15–15.1 min, 100–5% B; 15.1–16 min, 5% B. The mass spectrometry (MS) analysis was conducted under optimized electrospray ionization (ESI) conditions. The mass range analyzed was *m/z* 100–1000, with spray voltages of 3800 V in positive ion mode and 3000 V in negative ion mode. The capillary temperature was maintained at 320 °C, and collision energies of 10, 20, and 40 eV were applied for fragmentation. Sheath gas and auxiliary gas flows (N_2_) were set at 35 and 8 arbitrary units, respectively, to ensure stable ionization and detection.

#### 4.4.3. Metabolomics Analysis

For metabolomics analysis, the raw data files were imported into Progqenesis QI software (version 2.0) (Waters Corporation, Milford, CT, USA) for preprocessing. This process generated a comprehensive matrix with retention times, mass/charge ratios, peak intensities, and other relevant parameters. Preprocessing included baseline filtering, peak identification, integration, retention time correction, peak alignment, and normalization, performed by Lu-Ming Biotech Co., Ltd. (Shanghai, China).

To visualize metabolic alterations among treatment groups, we used principal component analysis (PCA) and partial least squares discriminant analysis (PLS-DA). Differential metabolites (DMs) were identified based on the variable importance in projection (VIP, >1) from PLS-DA, combined with the Student’s *t*-test (*p* < 0.05). We calculated the fold-change (FC) value for each metabolite by comparing the chronically fed YSD group with the CON group. For comprehensive pathway analyses, the DMs were annotated using the KEGG database. Mapping the DMs to the KEGG provided valuable pathway enrichment results, aiding in the exploration of underlying mechanisms.

### 4.5. Comprehensive Analysis of Transcriptome and Metabolome Data

Using the relative abundance data from both transcriptomes and metabolites, the top 100 DEGs and DMs were identified. To explore the associations between these genes and metabolites, the Pearson correlation algorithm was applied. Based on the results of the correlation analysis, heatmaps were generated to visually represent the strength and direction of these associations. A *p*-value threshold of <0.05 was used to select significant correlations, which were further visualized as a network diagram constructed with Cytoscape software (version 3.10.1, Cytoscape Consortium, San Diego, CA, USA). To better understand the interplay between DEGs and DMs, a KEGG Markup Language (KGML) network diagram was created using KGML files retrieved from the KEGG database. This diagram provided a comprehensive visual representation of the biological pathways linking the identified genes and metabolites.

### 4.6. Statistical Analysis

The Student’s *t*-test and one-way ANOVA were used for statistical comparisons. The data were presented as the mean ± SD, with *p* < 0.05 considered statistically significance. All statistical analyses were performed using GraphPad Prism 9.5 software (GraphPad Software, San Diego, CA, USA).

## 5. Conclusions

In summary, this study demonstrates that dietary supplementation with yeast culture may reduce the GR value of Small-Tail Han sheep without reducing body weight, and reducing the fat content of mutton. Specifically, we observed significant changes in immune-related gene expression and purine metabolite levels, which are associated with potential improvements in animal health and meat flavor. In addition, integrated transcriptomics and metabolomics analysis revealed that 1% yeast culture added to the diet may regulate related purine metabolites through the *AK4* and *RRM2* genes, contributing to enhanced meat quality (Figure 5). These findings provide valuable insights into the molecular mechanisms underlying the effects of yeast culture on meat quality and animal health.

## Figures and Tables

**Figure 1 ijms-26-04512-f001:**
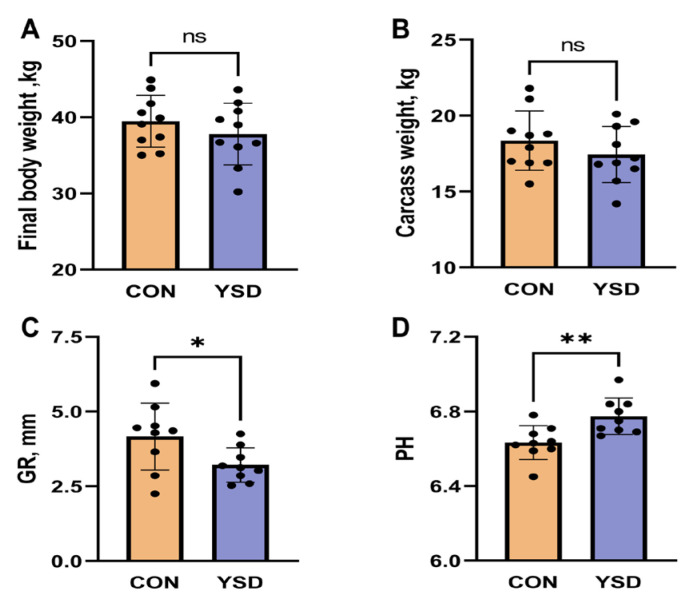
(**A**): Final body weight. (**B**): carcass weight, (**C**): girth rib (GR) value, (**D**): pH value of Small-tail Han sheep fed CON or YSD diet for 90 days. The data are presented as mean ± SD, n = 10. * indicates *p* < 0.05, ** indicates *p* < 0.01, ns indicates *p* > 0.05. CON, basic diet; YSD, basic diet supplemented with 1% yeast culture.

**Figure 2 ijms-26-04512-f002:**
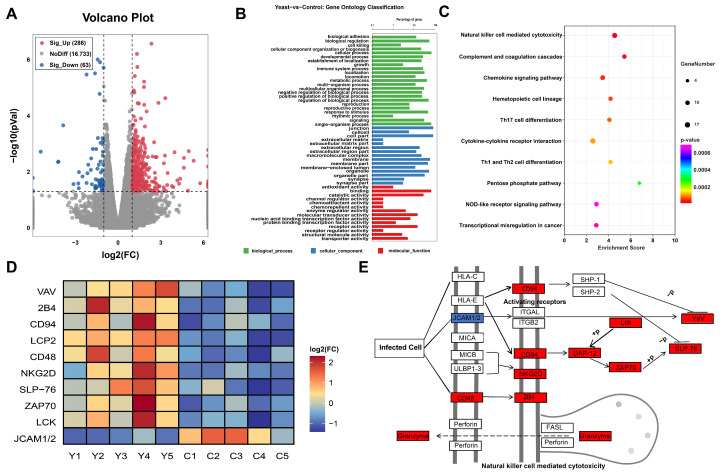
(**A**): Volcano plots, (**B**): GO enrichment, (**C**): KEGG pathway, (**D**): heatmap, (**E**): signaling cascade plot analysis of the DEGs in the LOD of Small-tail Han sheep fed CON or YSD diet for 90 days. DEGs, differentially expressed genes; GO, Gene Ontology; KEGG, Kyoto Encyclopedia of Genes and Genomes; LOD, Longissimus dorsi; CON, basic diet; YSD, basic diet supplemented with 1% yeast culture.

**Figure 3 ijms-26-04512-f003:**
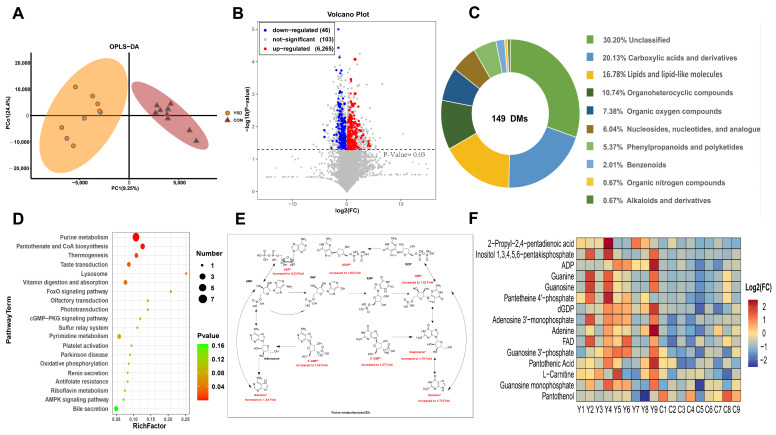
(**A**): OPLS-DA, (**B**): volcano plot, (**C**): pie chart, (**D**): KEGG pathway, (**E**): purine metabolism, (**F**): heatmap analysis of the DMs in the LOD of Small-tail Han sheep fed CON or YSD diet for 90 days. DMs, differential metabolites; KEGG, Kyoto Encyclopedia of Genes and Genomes; LOD, Longissimus dorsi; CON, basic diet; YSD, basic diet supplemented with 1% yeast culture.

**Figure 4 ijms-26-04512-f004:**
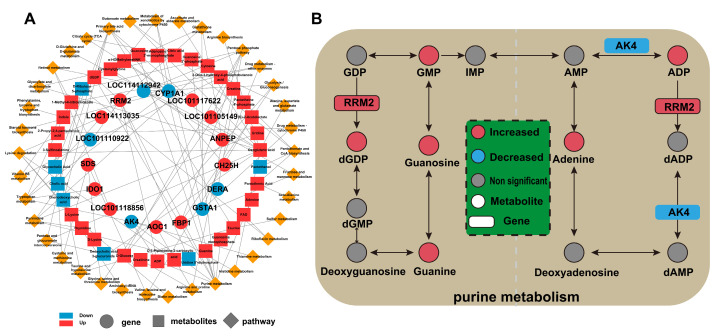
(**A**): Combined transcriptomics and metabolomics analysis in the LOD of Small-tail Han sheep fed CON or YSD diet for 90 days. (**A**): Correlation analysis of the top 100 DEGs and top 100 DMs. Circles represent DEGs, squares represent DMs, and small orange diamonds represent the pathway. Red indicates upregulation, while blue indicates downregulation. DEGs, differentially expressed genes; DMs, differential metabolites. (**B**): Association network diagram of *AK4*, *RRM2* and purine metabolites. Circles represent metabolites; rectangles represent genes; red indicates up-regulation; blue indicates downregulation; gray indicates non-significant. LOD, Longissimus dorsi; CON, basic diet; YSD, basic diet supplemented with 1% yeast culture.

**Figure 5 ijms-26-04512-f005:**
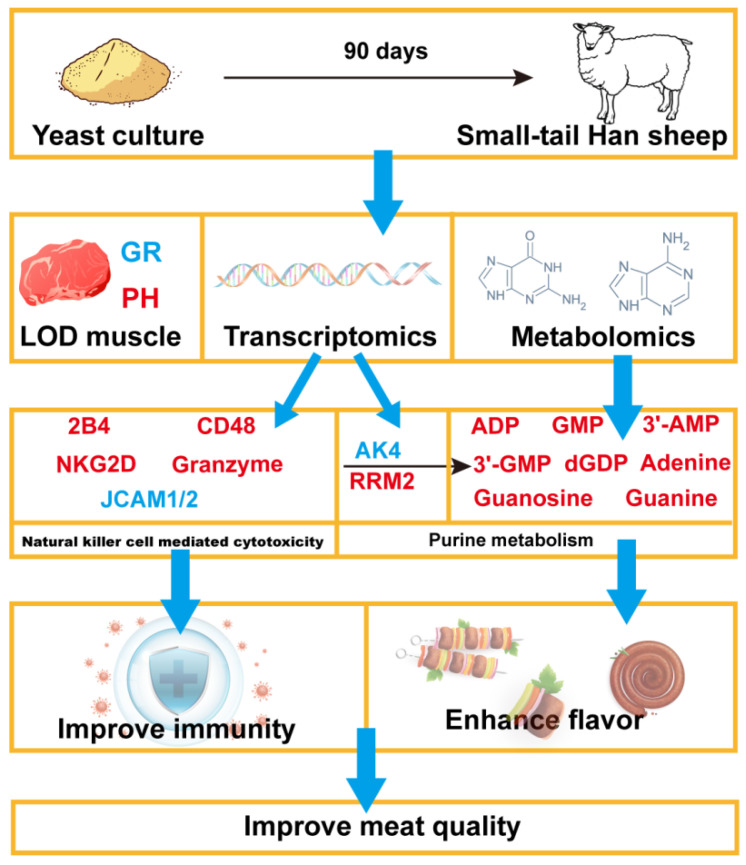
Scheme of postulated regulatory pathways for improvement of meat quality supplemented with yeast culture in Small-tail Han sheep. Red font indicates upregulation; blue font indicates downregulation.

**Table 1 ijms-26-04512-t001:** Ingredients and chemical composition of basal diets.

Ingredients, %	Content
Corn	56.00
Corn stalk	20.00
Soybean meal	5.60
Distiller’s Dried Grain with Solubles (corn)	5.20
Cottonseed meal	8.00
NaHCO_3_	0.96
Salt	0.24
Premix ^1^	4.00
Total	100
Nutrient content, %	
Metabolizable energy ^2^, MJ/kg	9.08
Crude fat	1.24
Crude protein	14.29
Neutral detergent fiber	48.73
Acid detergent fiber	21.84
Calcium	0.81
Phosphorus	0.53

^1^ Premix provided the following per kilogram of basal diet: VA 1,000,000 IU, VD 5,000,000 IU, VE. 2000 IU, Nicotinic acid 1000 mg, Fe 25 mg, Cu 250 mg, Mn 2000 mg, Zn 4000 mg, I 50 mg, Se 50 mg, Co 40 mg. ^2^ Metabolizable energy was calculated value, and the rest was measured value.

## Data Availability

The raw data supporting the conclusions of this article will be made available by the authors on request.

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
