# Peer review of "Yeast Culture Supplementation Improves Meat Quality by Enhancing Immune Response and Purine Metabolism of Small-Tail Han Sheep (Ovis aries)"

_ijms, 2025, doi:10.3390/ijms26104512_

Round 1
Reviewer 1 Report
Comments and Suggestions for Authors
The paper deals with an important practical problem: how could meat quality of a kind of locally kept sheep (short-tail Han Sheep) improved by controling yeast culture activity. The practical problem has been studied by scientifically very actual methodology which makes possible the imminent use of the results in sheep-keeping activity. Some features of the manuscript, however do not correspond the required standards. After the correction of these points the paper could be published.
Advices for corrections:
(1) Abbreviations should be explained at their first appearance in the manuscript.
(2) The systematic Latin name of the sheeps studied (small-tail Han Sheep) is lacking. This should be added, for enabling the Reader to find contact between published data and the results of the present publication.
(3) There is a paragraph in the manuscript, which is empty (6. Patents), even more under this heading completely different data are listed. This should be corrected.
(4) In the References the dio numbers are lacking. These should be searched and reported.
(5) At references to books not only the editorial company, but also its site should be given.
(6) Journal names are written inconsequently, for example, once you find Journal of animal science, while somewhat later Journal of Animal Science appears. The second variant is correct. Several citations suffer this lack of consistence.
MAIN QUESTION: It was formerly observed that addition of yeast to the usual (standard) diet of Small-tali Han sheep has advantageous effect on the health, immunity and fertility of these animals. The main goal of the study was to check whether this treatment influences the meat quality. This is a good idea and in the case of positive answer the treatment has broader applicability and certainly economic benefits.
ORIGINALITY: The goal is new in the sector of meat production, however it uses a treatment, which rightfully could be suspected to have positive consequences. Such relation between starting knowledge/data and research goals is usual in research activity.
RELEVANT: The methods used by the Authors are relevant to the main goal of their research project.
GAP: The research was conducted by good scientific logic, gaps have not been found by the present Reviewer.
NOVELTY: Yes, the research was new in its sector. It was logically based on earlier observations, as already stated above.
METHODS: The experimental methods have been selected from the most modern available methodology. The meat quality was tested through t-tests, transcriptome and metabolome assays and conducted by statistically significant number of animals.
CONCLUSIONS vs. RESULTS. The conclusions have been drawn by acceptable argumentation and led to clear results (THAT IS: the treatment studied in the present research has a positive effect on the meat quality).
TABLES, FIGURES. The composition of tables and drawing of the figures was made clearly, the reader can follow their meaning.
REFERENCES: Some corrections at the References are necessary, these have been described in my earlier report.
Comments on the Quality of English LanguageAn Editorial control of English grammar and orthography could improve the quality of the manuscript, which, however is understandable also in the present form.
Reviewer 2 Report
Comments and Suggestions for Authors
Natural enhancecers of quality meat are broadly researched currently, as consumers are more focus on sustainable& natural solutions. Yeast are well known from their actions related to presence of bioactive compounds, Authors claim underlines idea, that effect of yeast supplemenation on meat quality wasnt studied yet - clear goal of the research. The research was conducted on sheeps (40 individuals), and based on dietary support for 90 days (1% yeast addition - weakness of the study, analysing just single concentration of the yeast). Ex vivo sample was longissimus dorsi muscle, and laboratory work included transcriptome and metabolome analysis.
Abstract is informative.
Introduction is very brief and can be improved, not fully showing the bacground of the research & current state of the art. Just gap in the knowledge is not enough to start research.
point 4.1 please write more about animals & methode of slaughter. Why 1% of yeast addition was chosen. What are the characteristics of this yeast product?
4.3 was it pooled sample?
Other part of M&M description are fine
correctly prepared figures and figures legends
line 191-192 with this preliminary study on 40 animals, from which 20 were analyzed it is too-far reaching conclusion.
The reviewer still is missing the direct confirmation of exact yeast effect on productivity, strong part of the discussion is dedicated to immunity effect. In discussion Authors confirming previously obtained results- please think about underlining newelty of the study.
Discussion part seems to be not finished.
Conclusions part - did authors evaluated animals health status, as you claim to obtain health improvement? did you make sensory analysis dedicated to flavor changes?
conclusions too-far reaching
manuscript still needs some work of the Authors.
